# Epistemic Uncertainty and Observation Noise with the Neural Tangent Kernel

## Abstract

Recent work has shown that training wide neural networks with gradient descent is formally equivalent to computing the mean of the posterior distribution in a Gaussian Process (GP) with the Neural Tangent Kernel (NTK) as the prior covariance and zero aleatoric noise [12]. In this paper, we extend this framework in two ways. First, we show how to deal with non-zero aleatoric noise. Second, we derive an estimator for the posterior covariance, giving us a handle on epistemic uncertainty. Our proposed approach integrates seamlessly with standard training pipelines, as it involves training a small number of additional predictors using gradient descent on a mean squared error loss. We demonstrate the proof-of-concept of our method through empirical evaluation on synthetic regression.

## 1 Introduction

Jacot et al. have studied the training of wide neural networks, showing that gradient descent on a standard loss is, in the limit of many iterations, formally equivalent to computing the posterior mean of a Gaussian Process (GP), with the prior covariance specified by the Neural Tangent Kernel (NTK) and with zero aleatoric noise. Crucially, this insight allows us to study complex behaviours of wide networks using Bayesian nonparametrics, which are much better understood.

We extend this analysis by asking two research questions. First, we ask if a similar equivalence exists in cases where we want to do inference for arbitrary values of aleatoric noise. This is crucial in many real-world settings, where measurement accuracy or other data-gathering errors mean that the information in our dataset is only approximate. Second, we ask if it is possible to obtain an estimate of the posterior covariance, not just the mean. Since the posterior covariance measures the epistemic uncertainty about predictions of a model, it is crucial for problems that involve out-of-distribution detection or training with bandit-style feedback.

We answer both of these research questions in the affirmative. Our posterior mean estimator takes the aleatoric noise into account by adding a simple squared norm penalty on the deviation of the network parameters from their initial values, shedding light on regularization in deep learning. Our covariance estimator can be understood as an alternative to existing methods of epistemic uncertainty estimation, such as dropout [7, 20], the Laplace approximation [6, 19], epistemic neural networks [18], deep ensembles [21, 14] and Bayesian Neural Networks [3, 13]. Unlike these approaches, our method has the advantage that it can approximate the NTK-GP posterior arbitrarily well.

**Contributions** We derive estimators for the posterior mean and covariance of an NTK-GP with non-zero aleatoric noise, computable using gradient descent on a standard loss. We evaluate our results empirically on a toy repression problem.

## 2 Preliminaries

**Gaussian Processes** Gaussian Processes (GPs) are a popular non-parametric approach for modeling distributions over functions [22]. Given a dataset of input-output pairs $\{(\mathbf{x}_i, y_i)\}_{i=1}^N$, a GP represents uncertainty about function values by assuming they are jointly Gaussian with a covariance structure

Submitted to Workshop on Bayesian Decision-making and Uncertainty, 38th Conference on Neural Information Processing Systems (BDU at NeurIPS 2024). Do not distribute.

defined by a kernel function $k(\mathbf{x}, \mathbf{x}')$. The GP prior is specified as $f(\mathbf{x}) \sim \mathcal{GP}(m(\mathbf{x}), k(\mathbf{x}, \mathbf{x}'))$, where $m(\mathbf{x})$ is the mean function and $k(\mathbf{x}, \mathbf{x}')$ is the kernel. Assuming $y_i \sim \mathcal{N}(f(\mathbf{x}), \sigma^2)$ and given new test points $\mathbf{x}'$, the posterior mean and covariance are given by:

$$\boldsymbol{\mu}_p(\mathbf{x}') = m(\mathbf{x}') + \mathbf{K}(\mathbf{x}', \mathbf{x})^\top (\mathbf{K}(\mathbf{x}, \mathbf{x}) + \sigma^2 \mathbf{I})^{-1} (\mathbf{y} - m(\mathbf{x})), \tag{1}$$

$$\boldsymbol{\Sigma}_p(\mathbf{x}') = \mathbf{K}(\mathbf{x}', \mathbf{x}') - \mathbf{K}(\mathbf{x}', \mathbf{x})^\top (\mathbf{K}(\mathbf{x}, \mathbf{x}) + \sigma^2 \mathbf{I})^{-1} \mathbf{K}(\mathbf{x}', \mathbf{x}), \tag{2}$$

where $\mathbf{K}(\mathbf{x}, \mathbf{x})$ is the covariance matrix computed over the training inputs, $\mathbf{K}(\mathbf{x}', \mathbf{x})$ is the covariance matrix between the test and training points, and $\sigma^2$ represents the aleatoric (or observation) noise.

**Neural Tangent Kernel.**    The Neural Tangent Kernel (NTK) characterizes the evolution of wide neural network predictions as a linear model in function space. Given a neural network function $f(\mathbf{x}; \theta)$ parameterized by $\theta$, the NTK is defined through the Jacobian $J(\mathbf{x}) \in \mathbb{R}^{N \times p}$, where $J(\mathbf{x}) = \frac{\partial f(\mathbf{x}; \theta)}{\partial \theta}$, $N$ is the number of data points and $p$ is the number of parameters. The NTK at two sets of inputs $\mathbf{x}$ and $\mathbf{x}'$ is given by:

$$\mathbf{K}(\mathbf{x}, \mathbf{x}') = J(\mathbf{x}) J(\mathbf{x}')^\top. \tag{3}$$

Interestingly, as shown by [12] the NTK converges to a deterministic kernel and remains constant during training in the infinite-width limit. We call a GP with the kernel (3) the NTK GP.

## 3   Method

We now describe our proposed process of doing inference in the NTK-GP. Our procedure for estimating the posterior mean is given in Algorithm 1, while the procedure for the covariance is given in Algorithm 2. Note that our process is scaleable because both algorithms only use gradient descent, rather than relying on a matrix inverse in equations (1) and (2). While Algorithm 2 relies on the computation of the partial SVD of the Jacobian, we stress that efficient ways of doing so exist and do not require ever storing the full Jacobian. We defer the details of the partial SVD to Appendix E. We describe the theory that justifies our posterior computation in sections 3.1 and 3.2. We defer the discussion of literature to Appendix A.

---

**Algorithm 1** Algorithm for Computing the Posterior Mean in the NTK-GP

---

**procedure** TRAIN-POSTERIOR-MEAN($x_i, y_i, \theta_0$)
  $\hat{y}_i \leftarrow y_i + f(x_i; \theta_0)$                      ▷ Shift the targets to get zero prior mean (Lemma 3.2).
  $L \leftarrow \frac{1}{N} \sum_{i=1}^N (\hat{y}_i - f(x_i; \theta))^2 + \beta_N ||\theta - \theta_0||_2^2$              ▷ Equation (4)
  minimize $L$ with gradient descent wrt. $\theta$ until convergence to $\theta^\star$
  **return** $\theta^\star$                                           ▷ Return the trained weights.
**end procedure**

**procedure** QUERY-POSTERIOR-MEAN($x_j', \theta^\star, \theta^0$)                      ▷ $j = 1, \ldots, J$
  **return** $f(x_1'; \theta^\star) - f(x_1'; \theta^0), \ldots, f(x_J'; \theta^\star) - f(x_1'; \theta^0)$
**end procedure**

---

### 3.1   Aleatoric Noise

**Gradient Descent Converges to the NTK-GP Posterior Mean**    We build on the work of [12] by focusing on the computation of the mean posterior in the presence of **non-zero aleatoric noise**. We show that optimizing a regularized mean squared error loss in a neural network is equivalent to computing the mean posterior of an NTK-GP with non-zero aleatoric noise. In the following Lemma, we prove that for a sufficiently long training process, the predictions of the trained network converge to those of an NTK-GP with aleatoric noise characterized by $\sigma^2 = N\beta_N$. This is a similar result to [11], but from a Bayesian perspective rather than a frequentist generalization bound. Furthermore, our proof (see Appendix B) focuses on explicitly solving the gradient flows for test and training data points in function space.

**Lemma 3.1.** *Consider a parametric model $f(x; \theta)$ where $x \in \mathcal{X} \subset \mathbb{R}^N$ and $\theta \in \mathbb{R}^p$, initialized under some assumptions with parameters $\theta_0$. Minimizing the regularized mean squared error loss*

with respect to $\theta$ to find the optimal set of parameters $\theta^*$ over a dataset $(\mathbf{x}, \mathbf{y})$ of size $N$, and with sufficient training time $(t \to \infty)$:

$$\theta^* = \underset{\theta \in \mathbb{R}^p}{\arg\min} \frac{1}{N} \sum_{i=1}^{N} (y_i - f(x_i; \theta))^2 + \beta_N \|\theta - \theta_0\|_2^2, \tag{4}$$

is equivalent to computing the mean posterior of a Gaussian process with non-zero aleatoric noise, $\sigma^2 = N\beta_N$, and the NTK as its kernel:

$$f(\mathbf{x}'; \theta_\infty) = f(\mathbf{x}'; \theta_0) + \mathbf{K}(\mathbf{x}', \mathbf{x})(\mathbf{K}(\mathbf{x}, \mathbf{x}) + N\beta_N \mathbf{I})^{-1}(\mathbf{y} - f(\mathbf{x}; \theta_0)). \tag{5}$$

**Zero Prior Mean** In many practical scenarios, it is desirable to start with zero prior mean rather than with a prior mean that corresponds to random network initialization. To accommodate this, we introduce a simple yet effective transformation of the data and the network outputs, to be applied together with 3.1. We summarize it into the following lemma (see Appendix B for proof):

**Lemma 3.2.** *Consider the computational process derived in Lemma 3.1. Define shifted labels $\tilde{\mathbf{y}}$ and predictions $\tilde{f}(\mathbf{x}; \theta_\infty)$ as follows::*

$$\tilde{\mathbf{y}} = \mathbf{y} + f(\mathbf{x}; \theta_0), \quad \tilde{f}(\mathbf{x}; \theta_\infty) = f(\mathbf{x}; \theta_\infty) - f(\mathbf{x}'; \theta_0).$$

*Using these definitions, the posterior mean of a zero-mean Gaussian process can be computed as:*

$$\tilde{f}(\mathbf{x}', \theta_\infty) = \mathbf{K}(\mathbf{x}', \mathbf{x})(\mathbf{K}(\mathbf{x}, \mathbf{x}) + N\beta_N \mathbf{I})^{-1}\mathbf{y}. \tag{6}$$

---

**Algorithm 2** Algorithm for Computing the Posterior Covariance in the NTK-GP

---

**procedure** TRAIN-POSTERIOR-COVARIANCE$(x_i, K, \theta_0)$    $\triangleright$ $K$ is the number of predictors
  $U, \Sigma \leftarrow$ PARTIAL-SVD$(J_{\theta_0}(\mathbf{x}), K)$    $\triangleright$ Partial SVD of the Jacobian - see appendix E.
  **for** $i = 1, \dots, K$ **do**
    $\theta_i^\star \leftarrow$ TRAIN-POSTERIOR-MEAN$(x_i, U_i)$    $\triangleright$ $U_i$ is the $i$-th column of $U$.
  **end for**
  **for** $i = 1, \dots, K'$ **do**    $\triangleright$ Setting $K' = 0$ often works well (see Appendix D).
    $\theta'^\star_i \leftarrow$ TRAIN-POSTERIOR-MEAN$(x_i, \epsilon_i)$    $\triangleright$ $\epsilon_i \sim \mathcal{N}(0, \sigma^2 I)$
  **end for**
  **return** $\Sigma, \theta_1^\star, \dots, \theta_K^\star, \theta'^\star_1, \dots, \theta'^\star_{K'}$
**end procedure**

**procedure** QUERY-POSTERIOR-COVARIANCE$(x'_j, \Sigma, \theta_i^\star, \theta'^\star_i, \theta_0)$     $\triangleright$ $j = 1, \dots, J$
$$P \leftarrow \begin{bmatrix} f(x'_1; \theta_1^\star) - f(x'_1; \theta_0) & \dots & f(x'_1; \theta_K^\star) - f(x'_1; \theta_0) \\ & \dots & \\ f(x'_J; \theta_1^\star) - f(x'_J; \theta_0) & \dots & f(x'_J; \theta_K^\star) - f(x'_1; \theta_0) \end{bmatrix}, \quad P' \leftarrow \begin{bmatrix} f(x'_1; \theta'^\star_1) - f(x'_1; \theta_0) & \dots & f(x'_1; \theta'^\star_{K'}) - f(x'_1; \theta_0) \\ & \dots & \\ f(x'_J; \theta'^\star_1) - f(x'_J; \theta_0) & \dots & f(x'_J; \theta'^\star_{K'}) - f(x'_1; \theta_0) \end{bmatrix}$$
  **return** $J(x')J(x')^\top - P\Sigma^2 P^\top - P'(P')^\top / K'$    $\triangleright$ The last term vanishes for $K' = 0$
**end procedure**

---

## 3.2 Estimating the Covariance

We now justify Algorithm 2 for estimating the posterior covariance. The main observation that allows us to derive our estimator comes from examining the term $\mathbf{K}(\mathbf{x}', \mathbf{x})^\top (\mathbf{K}(\mathbf{x}, \mathbf{x}) + \sigma^2 \mathbf{I})^{-1} \mathbf{K}(\mathbf{x}', \mathbf{x})$ in the posterior covariance formula (2). This is summarized in the following Proposition.

**Proposition 3.1.** *Diagonalize $\mathbf{K}(\mathbf{x}, \mathbf{x})$ so that $\mathbf{K}(\mathbf{x}, \mathbf{x}) = U\Lambda U^\top$. We have*

$$\mathbf{K}(\mathbf{x}', \mathbf{x})^\top (\mathbf{K}(\mathbf{x}, \mathbf{x}) + \sigma^2 \mathbf{I})^{-1} \mathbf{K}(\mathbf{x}', \mathbf{x}) = (MU)\Lambda(MU)^\top + \sigma^2 MM^\top.$$

*Here, $M = \mathbf{K}(\mathbf{x}', \mathbf{x})^\top (\mathbf{K}(\mathbf{x}, \mathbf{x}) + \sigma^2 \mathbf{I})^{-1}$.*

*Proof.* We can rewrite it as:

$$\mathbf{K}(\mathbf{x}', \mathbf{x})^\top (\mathbf{K}(\mathbf{x}, \mathbf{x}) + \sigma^2 \mathbf{I})^{-1} \mathbf{K}(\mathbf{x}', \mathbf{x}) =$$
$$\underbrace{\mathbf{K}(\mathbf{x}', \mathbf{x})^\top (\mathbf{K}(\mathbf{x}, \mathbf{x}) + \sigma^2 \mathbf{I})^{-1}}_{M} (\mathbf{K}(\mathbf{x}, \mathbf{x}) + \sigma^2 \mathbf{I}) \underbrace{(\mathbf{K}(\mathbf{x}, \mathbf{x}) + \sigma^2 \mathbf{I})^{-1} \mathbf{K}(\mathbf{x}', \mathbf{x})}_{M^\top}$$

Denoting the term $\mathbf{K}(\mathbf{x}', \mathbf{x})^\top (\mathbf{K}(\mathbf{x}, \mathbf{x}) + \sigma^2 \mathbf{I})^{-1}$ with $M$, this can be written as:

$$\mathbf{K}(\mathbf{x}', \mathbf{x})^\top (\mathbf{K}(\mathbf{x}, \mathbf{x}) + \sigma^2 \mathbf{I})^{-1} \mathbf{K}(\mathbf{x}', \mathbf{x}) = (MU)\Lambda(MU)^\top + \sigma^2 M M^\top.$$

$\square$

The proposition is useful because the matrix $M$ appears in equation (1). Hence the matrix multiplication $MU$ is equivalent to estimating the posterior mean using algorithm 1 where targets are given by the columns of the matrix $U$. Hence the term $(MU)\Lambda(MU)^\top$ can be computed by gradient descent. In order to derive a complete estimator of the covariance, we still need to deal with the term $\sigma^2 M M^\top$. We can either estimate this term by fitting random targets (which corresponds to setting $K' > 0$ in algorithm 2) or accept an upper bound on the covariance, setting $K' = 0$. We describe this in detail in Appendix D.

## 4 Experiment

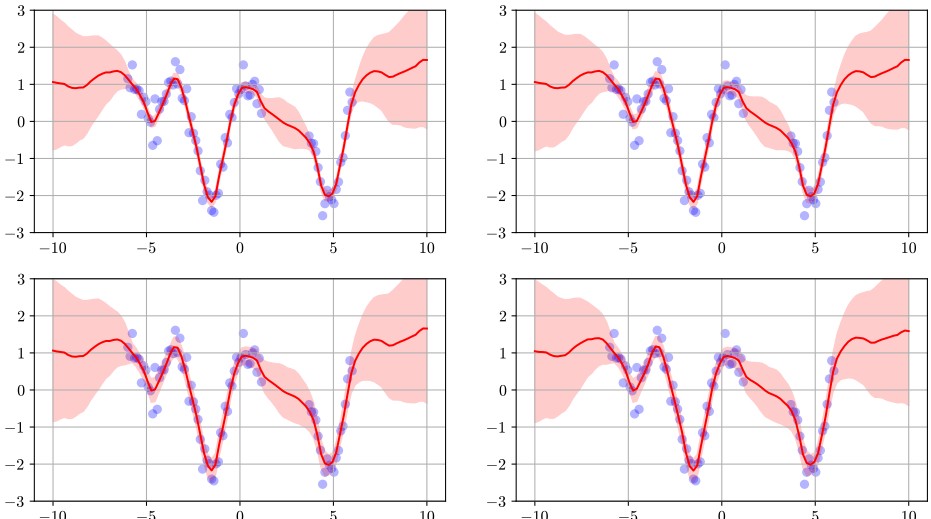

Figure 1: The NTK-GP posterior and its approximations: (top-left) Analytic Posterior, (top-right) Analytic upper bound on posterior (all eigenvectors), (bottom-left) Analytic upper bound on posterior (5 eigenvectors), (bottom-right) Posterior obtained with gradient descent ($K = 5$ predictors, $K' = 0$).

We applied the method to a toy regression problem shown in Figure 1. The problem is a standard non-linear 1d regression task which requires both interpolation and extrapolation. The top-left figure was obtained by computing the kernel of the NTK-GP using formula (3) and computing the posterior mean and covariance using equations (1) and (2). The top-right figure was obtained by analytically computing the upper bound defined in appendix D. The bottom-left figure was obtained by taking the first 5 eigenvectors of the kernel. Finally, the bottom-right figure was obtained by fitting a mean prediction network and 5 predictor networks using the gradient-descent method described in algorithm 2. The similarity of the figures shows that the method works. Details of network architecture are deferred to Appendix C.

## 5 Conclusions

This paper introduces a method for computing the posterior mean and covariance of NTK-Gaussian Processes with non-zero aleatoric noise. Our approach integrates seamlessly with standard training procedures using gradient descent, providing a practical tool for uncertainty estimation in contexts such as Bayesian optimization. The method has been validated empirically on a toy task, demonstrating its effectiveness in capturing uncertainty while maintaining computational efficiency. This work opens up opportunities for further research in applying NTK-GP frameworks to more complex scenarios and datasets.

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

## A  Related Work

**Neural Tangent Kernel**    The definition of the Neural Tangent Kernel (3), the proof of the fact that it stays constant during training and doesn't depend on initialization as well as the link to Gaussian Processes with no aleatoric noise are all due to the seminal paper [12]. The work of Lee et al. builds on that, showing that wide neural networks can be understood as linear models for purposes of studying their training dynamics, a fact we crucially rely on in the proof of our Lemma 3.1. Hu et al. describe a regularizer for networks trained in the NTK regime which leads to the same optimization problem used in our Lemma 3.1. The difference lies in the fact that we rely on the Bayesian interpretation of the network obtained at the end of training, while they focus on a frequentist generalization bound.

**Predictor Networks**    Prior work [17, 4, 5] has considered epistemic uncertainty estimation by fitting functions generated using a process that includes some kind of randomness. Burda et al. have applied a similar idea to reinforcement learning, obtaining exceptional results on Montezuma's Revenge, a problem where it is known that exploration very is hard. Ciosek et al. provided a link to Gaussian Processes, but did not leverage the NTK, instead describing an upper bound on a posterior relative to the kernel [16] where sampling corresponds to sampling from the network initialization. Osband et al. proposed[1] a way of sampling from a Bayesian linear regression posterior by solving an optimization problem with a similar structure to ours. However, this approach is different in two crucial ways. First, Osband et al. is interested in obtaining samples from the posterior, while we are interested in computing the posterior moments. Second, the sampling process in the paper by Osband et al. depends on the true regression targets in a way that our posterior covariance estimate does not. Also, our method is framed differently, as we intend it to be used in the context of the NTK regime, while Osband et al. discusses vanilla linear regression.

**Epistemic Uncertainty**    Our method of fitting the posterior covariance about network outputs can be thought of as quantifying epistemic uncertainty. There are several established methods in this space. Dropout [7, 20], works by randomly disabling neurons in a network and has a Bayesian interpretation. The Laplace approximation [6, 19] works by replacing an arbitrary likelihood with a Gaussian one. Epistemic neural networks [18] are based on the idea of using an additional input (the epistemic index) when training the network. Deep ensembles [21, 14] work by training several copies of a network with different initializations and sometimes training sets that are only partially overlapping. While classic deep ensembles do not have a Bayesian interpretation, He et al. have recently proposed a modification that approximates the posterior in the NTK-GP. Bayesian Neural Networks [3, 13] attempt to apply Bayes rule in the space of neural network parameters, applying various approximations. A full survey of methods of epistemic uncertainty estimation is beyond the scope of this paper.

## B  Proofs

**Lemma 3.1.** *Consider a parametric model $f(x; \theta)$ where $x \in \mathcal{X} \subset \mathbb{R}^N$ and $\theta \in \mathbb{R}^p$, initialized under some assumptions with parameters $\theta_0$. Minimizing the regularized mean squared error loss with respect to $\theta$ to find the optimal set of parameters $\theta^*$ over a dataset $(\mathbf{x}, \mathbf{y})$ of size $N$, and with sufficient training time $(t \to \infty)$:*

$$\theta^* = \underset{\theta \in \mathbb{R}^p}{\arg\min} \frac{1}{N} \sum_{i=1}^{N} (y_i - f(x_i; \theta))^2 + \beta_N ||\theta - \theta_0||_2^2, \tag{4}$$

*is equivalent to computing the mean posterior of a Gaussian process with non-zero aleatoric noise, $\sigma^2 = N\beta_N$, and the NTK as its kernel:*

$$f(\mathbf{x}'; \theta_\infty) = f(\mathbf{x}'; \theta_0) + \mathbf{K}(\mathbf{x}', \mathbf{x})(\mathbf{K}(\mathbf{x}, \mathbf{x}) + N\beta_N\mathbf{I})^{-1}(\mathbf{y} - f(\mathbf{x}; \theta_0)). \tag{5}$$

*Proof.* Consider a regression problem with the following regularized empirical loss:

$$\mathcal{L}(\mathbf{y}, f(\mathbf{x}; \theta)) = \frac{1}{N} \sum_{i=1}^{N} (y_i - f(x_i; \theta))^2 + \beta_N ||\theta - \theta_0||_2^2. \tag{7}$$

---

[1]See Section 5.3.1 in the paper by Osband et al.

Let us use $\theta_t$ to represent the parameters of the network evolving in time $t$ and let $\alpha$ be the learning rate. Assuming we train the network via continuous-time gradient flow, then the evolution of the parameters $\theta_t$ can be expressed as:

$$\frac{d\theta_t}{dt} = -\alpha \left[ \frac{2}{N} \nabla_\theta f(\mathbf{x}; \theta_t)(f(\mathbf{x}; \theta_t) - \mathbf{y}) + 2\beta_N(\theta_t - \theta_0) \right]. \tag{8}$$

Assuming that our neural network architecture operates in a sufficiently wide regime [15], where the first-order approximation remains valid throughout gradient descent, we obtain:

$$f(\mathbf{x}'; \theta_t) = f(\mathbf{x}'; \theta_0) + J_t(\mathbf{x}')(\theta_t - \theta_0) \rightarrow \nabla_\theta f(\mathbf{x}'; \theta_t)^\top = J_t(\mathbf{x}'). \tag{9}$$

The dynamics of the neural network on the training data:

$$\begin{aligned}
\frac{df(\mathbf{x}; \theta_t)}{dt} &= J_t(\mathbf{x})\frac{d\theta_t}{dt} = -\frac{2\alpha}{N} J_t(\mathbf{x}) \left[ J_t(\mathbf{x})^\top(f(\mathbf{x}; \theta_t) - \mathbf{y}) + \beta_N(\theta_t - \theta_0) \right] \\
&= -\frac{2\alpha}{N} \left( \mathbf{K}(\mathbf{x}, \mathbf{x})(f(\mathbf{x}; \theta_t) - \mathbf{y}) + \beta_N J_t(\mathbf{x})(\theta_t - \theta_0) \right) \\
&= -\frac{2\alpha}{N} \left( \mathbf{K}(\mathbf{x}, \mathbf{x})(f(\mathbf{x}; \theta_t) - \mathbf{y}) + \beta_N(f(\mathbf{x}; \theta_t) - f(\mathbf{x}; \theta_0)) \right) \\
&= -\frac{2\alpha}{N} \left( \mathbf{K}(\mathbf{x}, \mathbf{x}) + \beta_N \mathbf{I} \right) f(\mathbf{x}; \theta_t) + \frac{2\alpha}{N} \left( \mathbf{K}(\mathbf{x}, \mathbf{x})\mathbf{y} + \beta_N f(\mathbf{x}; \theta_0) \right)
\end{aligned}$$

This is a linear ODE, we can solve this:

$$\begin{aligned}
f(\mathbf{x}; \theta_t) = \ &\exp\left( -\frac{2\alpha}{N} t \left( \mathbf{K}(\mathbf{x}, \mathbf{x}) + \beta_N \mathbf{I} \right) \right) f(\mathbf{x}; \theta_0) \\
&- \frac{N}{2\alpha} \left( \mathbf{K}(\mathbf{x}, \mathbf{x}) + \beta_N \mathbf{I} \right)^{-1} \left[ \exp\left( -\frac{2\alpha}{N} t \left( \mathbf{K}(\mathbf{x}, \mathbf{x}) + \beta_N \mathbf{I} \right) \right) - \mathbf{I} \right] \\
&\times \frac{2\alpha}{N} \left( \mathbf{K}(\mathbf{x}, \mathbf{x})\mathbf{y} + \beta_N f(\mathbf{x}; \theta_0) \right)
\end{aligned}$$

Using $A^{-1}e^A = e^A A^{-1}$, and writing $\mathbf{K}(\mathbf{x}, \mathbf{x})y + \beta_N f(x, \theta_0) = (\mathbf{K}(\mathbf{x}, \mathbf{x}) + \beta_N I)f(x, \theta_0) + \mathbf{K}(\mathbf{x}, \mathbf{x})(y - f(x, \theta_0))$, we get:

$$\begin{aligned}
f(x, \theta_t) =\ &\exp\left( -\frac{2\alpha}{N} t(\mathbf{K}(\mathbf{x}, \mathbf{x}) + \beta_N I) \right) f(x, \theta_0) \\
&+ \left[ I - \exp\left( -\frac{2\alpha}{N} t(\mathbf{K}(\mathbf{x}, \mathbf{x}) + \beta_N I) \right) \right] (\mathbf{K}(\mathbf{x}, \mathbf{x}) + \beta_N I)^{-1}(\mathbf{K}(\mathbf{x}, \mathbf{x})y + \beta_N f(x, \theta_0)) \\
=\ &\exp\left( -\frac{2\alpha}{N} t(\mathbf{K}(\mathbf{x}, \mathbf{x}) + \beta_N I) \right) f(x, \theta_0) + \left[ I - \exp\left( -\frac{2\alpha}{N} t(\mathbf{K}(\mathbf{x}, \mathbf{x}) + \beta_N I) \right) \right] f(x, \theta_0) \\
&+ \left[ I - \exp\left( -\frac{2\alpha}{N} t(\mathbf{K}(\mathbf{x}, \mathbf{x}) + \beta_N I) \right) \right] (\mathbf{K}(\mathbf{x}, \mathbf{x}) + \beta_N I)^{-1}\mathbf{K}(\mathbf{x}, \mathbf{x})(y - f(x, \theta_0)) \\
=\ &f(x, \theta_0) + \left[ I - \exp\left( -\frac{2\alpha}{N} t(\mathbf{K}(\mathbf{x}, \mathbf{x}) + \beta_N I) \right) \right] (\mathbf{K}(\mathbf{x}, \mathbf{x}) + \beta_N I)^{-1}\mathbf{K}(\mathbf{x}, \mathbf{x})(y - f(x, \theta_0)).
\end{aligned}$$

Now, we consider the dynamics for the neural network of an arbitrary set of test points $\mathbf{x}'$:

$$\frac{df(x', \theta_t)}{dt} = -\frac{2\alpha}{N}\beta_N f(x', \theta_t) - \frac{2\alpha}{N} \left( \mathbf{K}(\mathbf{x}', \mathbf{x})(f(x, \theta_t) - y) - \beta_N f(x', \theta_0) \right). \tag{10}$$

222    This is a linear ODE with a time-dependent inhomogeneous term, we can solve it as follows:

$$f(x', \theta_t) = e^{-\frac{2\alpha}{N}\beta_N t} f(x', \theta_0) - \frac{2\alpha}{N} e^{-\frac{2\alpha}{N}\beta_N t} \int_0^t e^{\frac{2\alpha}{N}\beta_N u} \left(\mathbf{K}(x', \mathbf{x})(f(x, \theta_u) - y) - \beta_N f(x', \theta_0)\right) du$$

$$= e^{-\frac{2\alpha}{N}\beta_N t} f(x', \theta_0) + \frac{2\alpha}{N} e^{-\frac{2\alpha}{N}\beta_N t} \int_0^t e^{\frac{2\alpha}{N}\beta_N u} du \left(\mathbf{K}(x', \mathbf{x})y + \beta_N f(x', \theta_0)\right)$$

$$- \frac{2\alpha}{N} e^{-\frac{2\alpha}{N}\beta_N t} \mathbf{K}(x', \mathbf{x}) \int_0^t e^{\frac{2\alpha}{N}\beta_N u} f(x, \theta_u) du.$$

$$= e^{-\frac{2\alpha}{N}\beta_N t} f(x', \theta_0) + e^{-\frac{2\alpha}{N}\beta_N t} \frac{1}{\beta_N} \left(e^{\frac{2\alpha}{N}\beta_N t} - 1\right)(\mathbf{K}(x', \mathbf{x})y + \beta_N f(x', \theta_0))$$

$$- \frac{2\alpha}{N} e^{-\frac{2\alpha}{N}\beta_N t} \mathbf{K}(x', \mathbf{x}) \int_0^t e^{\frac{2\alpha}{N}\beta_N u} f(x, \theta_0) du$$

$$- \frac{2\alpha}{N} e^{-\frac{2\alpha}{N}\beta_N t} \mathbf{K}(x', \mathbf{x}) \int_0^t e^{\frac{2\alpha}{N}\beta_N u} \left[I - \exp\left(-\frac{2\alpha}{N}u(\mathbf{K}(\mathbf{x}, \mathbf{x}) + \beta_N I)\right)\right] du$$

$$\times (\mathbf{K}(\mathbf{x}, \mathbf{x}) + \beta_N I)^{-1} \mathbf{K}(\mathbf{x}, \mathbf{x})(y - f(x, \theta_0)).$$

$$= f(x', \theta_0) + \frac{1}{\beta_N}(1 - e^{\frac{2\alpha}{N}\beta_N t})\mathbf{K}(x', \mathbf{x})y - \frac{1}{\beta_N}(1 - e^{-\frac{2\alpha}{N}\beta_N t})\mathbf{K}(x', \mathbf{x})f(x, \theta_0)$$

$$- \frac{2\alpha}{N} e^{-\frac{2\alpha}{N}\beta_N t} \mathbf{K}(x', \mathbf{x}) \left[\frac{N}{2\alpha\beta}(e^{\frac{2\alpha}{N}\beta_N t} - 1)I - \frac{N}{2\alpha\beta}\mathbf{K}(\mathbf{x}, \mathbf{x})^{-1}\left(\exp\left(-\frac{2\alpha}{N}t\mathbf{K}(\mathbf{x}, \mathbf{x})\right) - I\right)\right]$$

$$\times (\mathbf{K}(\mathbf{x}, \mathbf{x}) + \beta_N I)^{-1} \mathbf{K}(\mathbf{x}, \mathbf{x})(y - f(x, \theta_0))$$

$$= f(x', \theta_0) + \frac{1}{\beta_N}(1 - e^{\frac{2\alpha}{N}\beta_N t})\mathbf{K}(x', \mathbf{x})(y - f(x, \theta_0))$$

$$- \frac{1}{\beta}\mathbf{K}(x', \mathbf{x}) \left[(1 - e^{-\frac{2\alpha}{N}\beta_N t})I - \mathbf{K}(\mathbf{x}, \mathbf{x})^{-1}\left(\exp\left(-\frac{2\alpha}{N}t(\mathbf{K}(\mathbf{x}, \mathbf{x}) + \beta_N I)\right) - e^{-\frac{2\alpha}{N}\beta_N t}I\right)\right]$$

$$\times (\mathbf{K}(\mathbf{x}, \mathbf{x}) + \beta_N I)^{-1} \mathbf{K}(\mathbf{x}, \mathbf{x})(y - f(x, \theta_0)).$$

223    Lastly, taking $t \to \infty$, we get

$$f(x', \theta_\infty) = f(x', \theta_0) + \frac{1}{\beta_N}\mathbf{K}(x', \mathbf{x})(y - f(x, \theta_0)) - \frac{1}{\beta_N}\mathbf{K}(x', \mathbf{x})(\mathbf{K}(\mathbf{x}, \mathbf{x}) + \beta_N I)^{-1}\mathbf{K}(\mathbf{x}, \mathbf{x})(y - f(x, \theta_0))$$

$$= f(x', \theta_0) + \frac{1}{\beta_N}\mathbf{K}(x', \mathbf{x})\left(I - (\mathbf{K}(\mathbf{x}, \mathbf{x}) + \beta_N I)^{-1}\mathbf{K}(\mathbf{x}, \mathbf{x})\right)(y - f(x, \theta_0))$$

$$= f(x', \theta_0) + \mathbf{K}(x', \mathbf{x})(\mathbf{K}(\mathbf{x}, \mathbf{x}) + \beta_N I)^{-1}(y - f(x, \theta_0)),$$

224    we achieve the desired result and hence having a regularized gradient flow in the infinite-width limit
225    is equivalent to inferring the mean posterior of a non-zero aleatoric noise NTK-GP.      □

226    **Lemma 3.2.** *Consider the computational process derived in Lemma 3.1. Define shifted labels $\tilde{\mathbf{y}}$ and*
227    *predictions $\tilde{f}(\mathbf{x}; \theta_\infty)$ as follows::*

$$\tilde{\mathbf{y}} = \mathbf{y} + f(\mathbf{x}; \theta_0), \quad \tilde{f}(\mathbf{x}; \theta_\infty) = f(\mathbf{x}; \theta_\infty) - f(\mathbf{x}'; \theta_0).$$

228    *Using these definitions, the posterior mean of a zero-mean Gaussian process can be computed as:*

$$\tilde{f}(\mathbf{x}', \theta_\infty) = \mathbf{K}(\mathbf{x}', \mathbf{x})(\mathbf{K}(\mathbf{x}, \mathbf{x}) + N\beta_N \mathbf{I})^{-1}\mathbf{y}. \tag{6}$$

229    *Proof.* Firstly, substituting $\tilde{\mathbf{y}}$ into $\mathbf{y}$:

$$f(\mathbf{x}'; \theta_\infty) = f(\mathbf{x}'; \theta_0) + \mathbf{K}(\mathbf{x}', \mathbf{x})(\mathbf{K}(\mathbf{x}, \mathbf{x}) + N\beta_N \mathbf{I})^{-1}(\tilde{\mathbf{y}} - f(\mathbf{x}; \theta_0))$$

$$= f(\mathbf{x}'; \theta_0) + \mathbf{K}(\mathbf{x}', \mathbf{x})(\mathbf{K}(\mathbf{x}, \mathbf{x}) + N\beta_N \mathbf{I})^{-1}\mathbf{y}$$

230    Now, using this new computational process, scaling it as $\tilde{f}(\mathbf{x}; \theta_\infty)$:

$$\tilde{f}(\mathbf{x}; \theta_\infty) = f(\mathbf{x}; \theta_\infty) - f(\mathbf{x}'; \theta_0) = \mathbf{K}(\mathbf{x}', \mathbf{x})(\mathbf{K}(\mathbf{x}, \mathbf{x}) + N\beta_N \mathbf{I})^{-1}\mathbf{y},$$

231    achieving the desired zero-mean Gaussian process.      □

## C   Details of the Experimental Setup

The Adam optimizer was used whenever our experiments needed gradient descent. A patience-based stopping rule was used where training was stopped if there was no improvement in the loss for 500 epochs. The other hyperparameters are given in the table below.

| hyperparameter | value |
|---|---|
| no of hidden layers | 2 |
| size of hidden layer | 512 |
| non-linearity | softplus |
| softplus beta | 87.09 |
| scaling multiplier in the output | 3.5 |
| learning rate for network predicting mean | 1e-4 |
| learning rate for covariance predictor networks | 5e-5 |

Moreover, we used trigonometric normalization, where an input point $x$ is first scaled and shifted to lie between 0 and $\pi$, obtaining a normalized point $x'$. The point $x'$ is then represented with a vector $[\sin(x'), \cos(x')]$.

## D   Details on Estimating The Covariance

We now describe two of dealing with the term $\sigma^2 M M^\top$ in the covariance formula. Upper bounding the covariance is described in Section D.1, while estimating the exact covariance by fitting noisy targets is described in Section D.2.

### D.1   Upper Bounding the Covariance

First, we can simply ignore the term in our estimator, obtaining an upper bound on the covariance. We now characterize the tightness of the upper bound, i.e. the magnitude of the term

$$\sigma^2 M M^\top = \sigma^2 \mathbf{K}(\mathbf{x}', \mathbf{x})(\mathbf{K}(\mathbf{x}, \mathbf{x}) + \sigma^2 \mathbf{I})^{-1}(\mathbf{K}(\mathbf{x}, \mathbf{x}) + \sigma^2 \mathbf{I})^{-1}\mathbf{K}(\mathbf{x}', \mathbf{x})^\top.$$

We do this is the following two lemmas.

**Lemma D.1.** *When* $\mathbf{x} = \mathbf{x}'$*, i.e. on the training set, we have*

$$\sigma^2 \mathbf{K}(\mathbf{x}', \mathbf{x})(\mathbf{K}(\mathbf{x}, \mathbf{x}) + \sigma^2 \mathbf{I})^{-1}(\mathbf{K}(\mathbf{x}, \mathbf{x}) + \sigma^2 \mathbf{I})^{-1}\mathbf{K}(\mathbf{x}', \mathbf{x})^\top \preccurlyeq \sigma^2 \mathbf{I}.$$

*Proof.* By assumption, $\mathbf{K}(\mathbf{x}', \mathbf{x}) = \mathbf{K}(\mathbf{x}, \mathbf{x}) = \mathbf{K}$. Denote the diagonalization of $\mathbf{K}$ with $\mathbf{K} = \mathbf{U}\mathbf{\Lambda}\mathbf{U}^\top$. We have

$$\begin{aligned}
&\sigma^2 \mathbf{K}(\mathbf{x}', \mathbf{x})(\mathbf{K}(\mathbf{x}, \mathbf{x}) + \sigma^2 \mathbf{I})^{-1}(\mathbf{K}(\mathbf{x}, \mathbf{x}) + \sigma^2 \mathbf{I})^{-1}\mathbf{K}(\mathbf{x}', \mathbf{x})^\top \\
&= \sigma^2 \mathbf{K}(\mathbf{K} + \sigma^2 \mathbf{I})^{-2}\mathbf{K}^\top \\
&= \sigma^2 \mathbf{U}\mathbf{\Lambda}\mathbf{U}^\top(\mathbf{U}\mathbf{\Lambda}\mathbf{U}^\top + \sigma^2 \mathbf{I})^{-2}\mathbf{U}\mathbf{\Lambda}\mathbf{U}^\top \\
&= \sigma^2 \mathbf{U}\mathbf{\Lambda}\mathbf{U}^\top\mathbf{U}(\mathbf{\Lambda} + \sigma^2 \mathbf{I})^{-2}\mathbf{U}^\top\mathbf{U}\mathbf{\Lambda}\mathbf{U}^\top \\
&= \sigma^2 \mathbf{U}\mathbf{\Lambda}(\mathbf{\Lambda} + \sigma^2 \mathbf{I})^{-2}\mathbf{\Lambda}\mathbf{U}^\top.
\end{aligned}$$

It can be seen that the diagonal entries of $\mathbf{\Lambda}(\mathbf{\Lambda} + \sigma^2 \mathbf{I})^{-2}\mathbf{\Lambda}$ are less than or equal one. $\square$

The Lemma above, stated in words, implies that, on the training set, the variance estimates that come from using the upper bound (which doesn't require us to fit noisy targets as in Section D.2) are off by at most $\sigma^2$.

We now give another Lemma, which characterizes the upper bound on arbitrary test points, not just the training set.

**Lemma D.2.** *Denote by* $\lambda_{max}$ *the maximum singular value of* $\mathbf{K}(\mathbf{x}', \mathbf{x}')$*. Then we have*

$$\left\|\sigma^2 \mathbf{K}(\mathbf{x}', \mathbf{x})(\mathbf{K}(\mathbf{x}, \mathbf{x}) + \sigma^2 \mathbf{I})^{-1}(\mathbf{K}(\mathbf{x}, \mathbf{x}) + \sigma^2 \mathbf{I})^{-1}\mathbf{K}(\mathbf{x}', \mathbf{x})^\top\right\|_2 \leq \frac{1}{4}\lambda_{max}.$$

*Proof.* By Proposition 1.3.2 from the book by Bhatia, we have that

$$\mathbf{K}(\mathbf{x}', \mathbf{x})^\top = \mathbf{K}(\mathbf{x}, \mathbf{x})^{1/2} \mathbf{C} \mathbf{K}(\mathbf{x}', \mathbf{x}')^{1/2},$$

where $\mathbf{C}$ is a contraction. Denote the diagonalization of $\mathbf{K}(\mathbf{x}, \mathbf{x})$ with $\mathbf{K}(\mathbf{x}, \mathbf{x}) = \mathbf{U}\mathbf{\Lambda}\mathbf{U}^\top$. We have

$$
\begin{aligned}
& \left\| \sigma^2 \mathbf{K}(\mathbf{x}', \mathbf{x})(\mathbf{K}(\mathbf{x}, \mathbf{x}) + \sigma^2 \mathbf{I})^{-2} \mathbf{K}(\mathbf{x}', \mathbf{x})^\top \right\|_2 \\
=& \left\| \sigma^2 \mathbf{K}(\mathbf{x}', \mathbf{x}')^{1/2} \mathbf{C}^\top \mathbf{K}(\mathbf{x}, \mathbf{x})^{1/2} (\mathbf{K}(\mathbf{x}, \mathbf{x}) + \sigma^2 \mathbf{I})^{-2} \mathbf{K}(\mathbf{x}, \mathbf{x})^{1/2} \mathbf{C} \mathbf{K}(\mathbf{x}', \mathbf{x}')^{1/2} \right\|_2 \\
\leq& \sigma^2 \lambda_{\max} \left\| \mathbf{K}(\mathbf{x}, \mathbf{x})^{1/2} (\mathbf{K}(\mathbf{x}, \mathbf{x}) + \sigma^2 \mathbf{I})^{-2} \mathbf{K}(\mathbf{x}, \mathbf{x})^{1/2} \right\|_2 \\
=& \sigma^2 \lambda_{\max} \left\| \mathbf{U}\mathbf{\Lambda}^{1/2}\mathbf{U}^\top \mathbf{U}(\mathbf{\Lambda} + \sigma^2 \mathbf{I})^{-2} \mathbf{U}^\top \mathbf{U}\mathbf{\Lambda}^{1/2}\mathbf{U}^\top \right\|_2 \\
=& \sigma^2 \lambda_{\max} \left\| \mathbf{\Lambda}^{1/2}(\mathbf{\Lambda} + \sigma^2 \mathbf{I})^{-2} \mathbf{\Lambda}^{1/2} \right\|_2.
\end{aligned}
$$

We can expand $\left\| \mathbf{\Lambda}^{1/2}(\mathbf{\Lambda} + \sigma^2 \mathbf{I})^{-2} \mathbf{\Lambda}^{1/2} \right\|_2$ as $\max_i \left\{ \frac{\lambda_i}{(\lambda_i + \sigma^2)^2} \right\} \leq \frac{1}{4\sigma^2}$, which gives the desired result. $\qquad\square$

## D.2 Exact Covariance by Fitting Noisy Targets

In certain cases, we might not be satisfied with having an upper bound on the posterior covariance, even if it is reasonably tight. We can address these scenario by fitting additional predictor networks, trained on targets sampled from the spherical normal. Formally, we have

$$\sigma^2 M M^\top = M \mathbb{E}_\epsilon \left[ \epsilon \epsilon^\top \right] M^\top,$$

where $\epsilon \sim \mathcal{N}(0, \sigma^2 I)$. We can take $K'$ samples $\epsilon_1, \ldots, \epsilon_{K'}$, obtaining

$$M \mathbb{E}_\epsilon \left[ \epsilon \epsilon^\top \right] M^\top \approx \frac{1}{K'} \sum_i M \epsilon_i \epsilon_i^\top M^\top = \frac{1}{K'} \sum_i (M \epsilon_i)(M \epsilon_i)^\top, \tag{11}$$

where the approximation becomes exact by the law of large numbers as $K' \to \infty$. Since the multiplication $M \epsilon_i$ is equivalent to estimating the posterior mean with algorithm 1, we can perform the computation in equation (11) by gradient descent.

# E Computing The Partial SVD

Our Algorithm 2 includes the computation of the partial SVD of the Jacobian:

$$U, \Sigma \leftarrow \text{Partial-SVD}(J_{\theta_0}(\mathbf{x}), K).$$

We require an SVD which is partial in the sense that we only want to compute the first $K$ singular values. For the regression experiment in this submission, we simply called the full SVD on the Jacobian and took the first $K$ columns of $U$ and the first $K$ diagonal entries of $\Sigma$. This process is infeasible for larger problem instances.

This can be addressed by observing that the power method for SVD computation [2] only requires computing Jacobian-vector products and vector-Jacobian products, which can be efficiently computed in deep learning frameworks without access to the full Jacobian. Another approach that avoids constructing the full Jacobian is the use of randomized SVD [9]. We leave the implementation of these ideas to further work.

# F Network Initialization

We consider a neural network model $f(x; \theta)$, where $\theta \in \mathbb{R}^p$ denotes the set of parameters. The model consists of $L$ layers with dimensions $\{n_0, n_1, \ldots, n_L\}$, where $n_0$ is the input dimension and $n_L$ is the output dimension. Note that, as we want to leverage the theory of wide networks, the number of neurons in the hidden layers, $\{n_2, \ldots, n_{L-1}\}$, is large.

For each fully connected layer $l$, the weight matrix $W^{(l)} \in \mathbb{R}^{n_l \times n_{l-1}}$ and the bias vector $b^{(l)} \in \mathbb{R}^{n_l}$ are initialized from a Gaussian distribution with mean zero and standard deviations $\sigma_w$ and $\sigma_b$, respectively:

$$W_{ij}^{(l)} \sim \mathcal{N}(0, \sigma_w^2), \quad b_j^{(l)} \sim \mathcal{N}(0, \sigma_b^2),$$

where $\sigma_w$ and $\sigma_b$ are fixed values set as hyperparameters during initialization (we use $\sigma_w = \sigma_b = 1$).

The network uses a non-linear activation function $\sigma : \mathbb{R} \to \mathbb{R}$ with bounded second derivative, ensuring Lipschitz continuity. The output of each layer $l$ is scaled by $1/\sqrt{n_l}$ to maintain the appropriate magnitude, particularly when considering the infinite-width limit:

$$a^{(l)} = \sigma \left( \frac{1}{\sqrt{n_l}} W^{(l)} a^{(l-1)} + b^{(l)} \right),$$

where $a^{(l)}$ is the output of layer $l$, and $a^{(0)} = x$ is the input to the network.

The final layer output is further scaled by a constant factor $c_{\text{out}}$ to ensure that the overall network output remains within the desired range. Specifically, the output $f(x; \theta)$ is given by:

$$f(x; \theta) = \frac{c_{\text{out}}}{\sqrt{n_L}} W^{(L)} a^{(L-1)},$$

where $c_{\text{out}}$ is a predefined constant that ensures the final output is of the appropriate scale. In our model, $c_{\text{out}}$ is set to 3.5. For the hidden layers, we choose $\sigma(\cdot)$ to be Softplus − a smoothed version of ReLU. In this case, an additional scaling factor $\beta$ is introduced to modulate the sharpness of the non-linearity:

$$a^{(l)} = \sigma \left( \frac{1}{\sqrt{n_l}} W^{(l)} a^{(l-1)} + b^{(l)}; \beta \right).$$

In our model, we set $\beta = 87.09$ for the Softplus activation to ensure the appropriate range of activation values. The process described above is standard. We followed closely the methodology provided in several works in the literature [12][15][8]. This initialization strategy ensures that the network's activations and gradients do not explode or vanish as the number of neurons $n_l$ increases.

