# OpenReview forum: "Epistemic Uncertainty and Observation Noise with the Neural Tangent Kernel"
_NeurIPS.cc/2024/Workshop/BDU — Submitted to NeurIPS BDU Workshop 2024_

### Official Review · Reviewer_KHfd · 2024-09-30

**Rating:** 1
**Confidence:** 4

**Review:**

This paper studies NTK approximations of neural networks trained with a ridge penalty (modeling aleatoric noise, in the words of the authors) and attempt to estimate the posterior variance that corresponds to the NTK mean.

To the best of my knowledge, the authors' work has not only been investigated in the prior literature, but the authors' claims are also incorrect.

First, the author's proof that training a regularized neural network converges to a regularized NTK is catastrophically incomplete. This proof includes the line "assuming that our neural network architecture operates in a sufficiently wide regime [15], where the
first-order approximation remains valid throughout gradient descent, we obtain..." - this first-order approximation is only proven to hold when models are trained without regularization, at least according to the proofs of Jacot et al and Lee et al.

The rest of the authors' proof is then essentially trivial, as it is just amounts to solving an ODE. Proving that the NTK limit holds under regularization requires the much bigger step of proving that the first order approximation holds when training with regularizaiotn.

Second, the authors' estimate of variance is essentially the empirical estimate of variance from training multiple neural networks from different initializations. Lee et al (2019) prove that this is not an estimate of posterior variance and [He et al (2020](https://arxiv.org/abs/2007.05864) show how to actually obtain posterior variance estimates.

Much of the authors' lemmas also essentially amount to basic properties of SVD and shrinkage estimators that could be found in any introductory machine learning textbook.

---

### Official Review · Reviewer_PuqG · 2024-10-07

**Rating:** 4
**Confidence:** 3

**Review:**

Summary
------
Computing mean and covariance of the posterior distribution of a Gaussian Process with the neural tangent kernel is addressed, allowing for aleatoritc noise.

Main review
-------
The proposed method seems sensible, however, the presentation is not very rigorous and I don't consider the key theoretical justification of this method - lemma 3.1. - enough.

It is never said that $\beta_N > 0$ should be positive parameters or something similar. Also, the relevant assumptions on the model $f$ are missing, which at least include differentiability with respect to the second argument. In section 2, the relevant spaces (i.e. domains and codomains) should be stated, i.e. that (apparently) $\mathbf{x} \in \mathcal{X} \subset \mathbb{R}^N$, $y_i \in \Theta \subset \mathbb{R}^p$, $f \colon \mathcal{X} \times \Theta \to \mathbb{R}$.
  Should it instead say $y_i \sim \mathcal{N}(f(x_i), \sigma^2)$ perhaps? I would also be beneficial in my opinion to introduce notation for the training set and the test set. Is $J(x)$ the Jacobian for $N$ test points or why does it have the dimension $N \times p$ at every $x \in \mathbb{R}^N$ (assuming that $f$ is a scalar function)?
My main issue, however, is with its proof: a main assumption of the proof (this formulation alone should be a red flag) is that the network is trained with a continuous-time gradient flow, which clearly deviates from what happens in practice. The discretization error of the explicit discretization is not addressed, but vital for statements about what happens in practice. Furthermore, on one hand, the statement fixes a $N \in \mathbb{N}$ but on the other hand, it is assumes that the "neural network architecture operates in a sufficiently wide regime", which I understand as $N = \infty$ or that, up to some correction terms - which would need to be addressed - $N$ is sufficiently large. The notation $\rightarrow$ in equation (9) is not clear, is this meant as $\theta_t \to \theta_0$, i.e. for small $t > 0$? In the calculation after l. 217 I think it should be $N \beta_N (\theta_t - \theta_0)$, since the $\frac{1}{N}$ is pulled outside of the sum. This would then carry through the whole calculation.

The algorithms should include, in a transparent way, the input and output. All the calculations seems to be correct, up to what I mentioned above, however, I think that equation (10) deserves one sentence more of explanation.
The experiments section is very short and the plot should be labelled better, i.e. clearly say what the blue points and the red curve represent. The sentence "The similarly of the figures shows that the method works" is very meagre and not very convincing. Please ameliorate the discussion by giving explanations of why the plots different and how these differences come about. Furthermore, the choice of hyperparameters (e.g. $K = 5$) is not motivated at all.
In the conclusion "computational efficiency" is claimed, but there are no indications or comparisons in that direction anywhere in the manuscript.
Lastly, the code for reproducing the experiments should be made available publically.

References
-------
The bibliography is pretty sloppy. Among other things, capitalization is very inconsistent (i.e. the spelling of NeuRIPS or ICML, names such as "Bayes" are not capitalized sometimes, as is the book title of [2]), reference [4] has been published at ICML, "Vol. 2.3." in reference [22] seems wrong, and the way arxiv preprints and contributions to conferences are mentioned is also inconsistent. I would definitely remove the google books link from the first reference. Moreover, I recommend using the package "cite" so that when citing multiple references at once, their numbers are ordered automatically.
In the last paragraph of appendix A I recommend augmenting the last sentence by something like "..., and we refer to [...]", where some overview literature is cited.

In the introduction, it is not clear to me what is meant by "shedding light on regularization in deep learning" in l. 26, and would appreciate some elaboration.

Minor stuff
------
In l. 199 I would instead write "classical" instead of "classic".
In l. 187 it should be "are" instead of "is".
Before l. 221 the $=$ is wrongly indented, the same for the two $=$'s before l. 223.
l. 251: if I am not mistaken, the eigenvalues of $\Lambda (\Lambda + \sigma^2 I)^{-2} \Lambda$ are $\frac{\lambda_i^2}{\lambda_i + \sigma^2}$, which are even less than one, since $\sigma^2 > 0$. (However, if $\sigma^2 = 0$ is permissible is never mentioned.)
Since I am no expert, I would appreciate a reference for the statement that Jacboain-vector products can be efficiently computed in deep learning frameworks without access to the full Jacobian (ll. 278-279).
The round braces in l. 298 are unnecessary and this should also be changed in ll. 38 - 39.
The notation $\mathcal{GP}$ is never defined. Either put it in words or define it rigorously.
I think the formulations "asking two research questions" and "answer both these research questions in the affirmative" is peculiar, I would rather delete these two sentences.
Writing vectors in bold is not carried out consistently throughout, for example, in the statement of Lemma 3.1. $x$ is not bold, but sometimes in proof. Also, $y$ is not bold, but then it is the subsequent lemma.
The formula after l. 217 is missing a full stop at the end.
In the statement of proposition 3.1 I would instead write "$M \coloneqq$".
In the proof of Lemma 3.1, the notation $J_t \coloneqq \frac{\partial}{\partial \theta} \mid_{\theta = \theta_t} f(x, \theta)$ should be introduced to avoid confusion.
l. 292: please clearly state what is Lipschitz continuous here.
In l. 287, is $\ell \in \{ n_1, \ldots, n_{L - 1} \}$?
In the statement of Lemma 3.1 I would add the reference to equation (7) so that it is clear what "regularized mean squared error" refers to.

---

### Decision · Program_Chairs · 2024-10-09

**Decision:**

Reject

**Comment:**

One of the reviewers, who is a seasoned expert on the topic, raises issues with correctness of the paper's theory. If the reviewer is right, rejection is justified on that basis alone. If the reviewer is wrong due to a mistake on their part, rejection is also justified, as it indicates the work was seriously misunderstood by an expert and therefore its presentation should be made more clear. In either case, the paper is not ready for presentation, so I encourage the authors to improve on these issues and resubmit to a future workshop.